# Photoluminescence and Electrical Properties of n-Ce-Doped ZnO Nanoleaf/p-Diamond Heterojunction

**DOI:** 10.3390/nano12213773

**Published:** 2022-10-26

**Authors:** Qinglin Wang, Yu Yao, Xianhe Sang, Liangrui Zou, Shunhao Ge, Xueting Wang, Dong Zhang, Qingru Wang, Huawei Zhou, Jianchao Fan, Dandan Sang

**Affiliations:** 1Shandong Key Laboratory of Optical Communication Science and Technology, School of Physics Science and Information Technology, Liaocheng University, Liaocheng 252000, China; 2Shandong Liaocheng Laixin Powder Materials Science and Technology Co., Ltd., Liaocheng 252000, China; 3Ulsan Ship and Ocean College, Ludong University, Yantai 264000, China; 4School of Chemistry and Chemical Engineering, Liaocheng University, Liaocheng 252000, China

**Keywords:** Ce-doped ZnO NLs, boron-doped diamond, heterojunction, photoluminescence, electrical transport behavior, high temperature

## Abstract

The n-type Ce:ZnO (NL) grown using a hydrothermal method was deposited on a p-type boron-doped nanoleaf diamond (BDD) film to fabricate an n-Ce:ZnO NL/p-BDD heterojunction. It shows a significant enhancement in photoluminescence (PL) intensity and a more pronounced blue shift of the UV emission peak (from 385 nm to 365 nm) compared with the undoped heterojunction (n-ZnO/p-BDD). The prepared heterojunction devices demonstrate good thermal stability and excellent rectification characteristics at different temperatures. As the temperature increases, the turn-on voltage and ideal factor (*n*) of the device gradually decrease. The electronic transport behaviors depending on temperature of the heterojunction at different bias voltages are discussed using an equilibrium band diagram and semiconductor theoretical model.

## 1. Introduction

Nanomaterials (nanowires, nanorods, nanotubes, nanospheres, etc.) have become a key research target for researchers due to their potential applications in transistors, light-emitting diodes (LEDs), and sensing devices [1]. Nanostructured ZnO with unique properties (wide band gap (3.3 eV), large saturation velocity, large exciton binding energy (60 meV), etc.) is widely used to fabricate various nanodevices, including Schottky diodes, photodetectors, chemical sensors, and surface acoustic wave devices [2,3,4,5]. It is considered one of the attractive semiconductors in the field of nanoscale electronic and photonic devices. Various ZnO nanostructures have been fabricated into p–n heterojunctions with excellent properties on numerous substrates (such as Si [6,7], SiC [8,9], CuO [10], and graphene [11]). Compared with the conventional substrate materials above, boron-doped diamond (BDD) has outstanding properties, such as wide forbidden band (5.47 eV), chemical stability, high thermal conductivity, high electron–hole mobility (2400 cm^2^/(V · s)), high breakdown voltage (~10^7^ V/cm), and radiation resistance, so it is considered a new type of high-temperature-resistant p-type semiconductor material. To combine the outstanding properties of ZnO and BDD, ZnO–BDD-related devices could be prepared to extend the applications in optoelectronics, especially in harsh environments, such as high radiation and temperature. In recent years, related work on the successful combination of p-type BDD with n-type ZnO to fabricate heterojunctions has been reported [12,13,14,15,16,17], and typical rectification characteristics and/or negative differential resistance at high temperature have been exhibited. However, the optical and electrical properties of the heterojunction system should be improved further for high stability in optoelectronic field applications.

In order to enhance the optoelectronic properties of n-ZnO/p-BDD heterojunction systems, high-quality p–n junctions have to be fabricated, and higher-quality n-type ZnO semiconductors are required. Doping is a viable method to improve the quality and performance of semiconductors. Rare earth (RE) doping of ZnO is very interesting and attractive because the shielded 4f energy level of Re^3+^ can cause multifarious clearly defined narrow optical jumps between the spin–orbit energy level splitting in different manifold weak crystal fields, thus causing significant changes in the optoelectronics of the semiconductor [18]. There have been some reports on RE elements doped with ZnO, including Er, La, Yb, and Eu [19,20,21]. Among the RE elements, Ce is of interest due to its unique property of readily forming oxygen vacancies and controlling the morphology of the host, and the adaptability of Ce-doped ZnO has been demonstrated [22,23,24]. Up to now, there has been no comparative study of the electronic and optical properties of Ce-doped nanostructured n-ZnO/p-BDD heterojunctions. In this work, Ce-doped ZnO nanoleaves (NLs) were deposited on p-BDD films by using a hydrothermal method, and the photoluminescence (PL) properties of the n-Ce-doped ZnO NL/p-BDD heterojunction were investigated in detail. The temperature-related electrical performance carrier transport behavior was analyzed.

## 2. Results and Discussion

Figure 1a–c shows the scanning electron microscopy (SEM) of the BDD substrate and the Ce-doped ZnO NLs grown on the BDD. In Figure 1a, the BDD films are mainly composed of dense diamond particles with an estimated average dimension of 1.4 μm. With the doping of Ce ions in the ZnO crystal lattice, the formation of NL-like and clearly identified three-dimensional morphology clusters is observed (Figure 1b,c). They are mainly composed of NLs with an average length (thickness) of 1.6 μm (7.9 nm) interspersed longitudinally.

Typical X-ray diffraction (XRD) peaks of the n-Ce:ZnO NL/p-BDD heterojunction are given in Figure 1d. The peaks located at 31.73°, 34.42°, 36.22°, 47.55°, 56.59°, 62.82°, 67.99°, and 69.07° can be labeled as the (100), (002), (101), (102), (110), (103), (112), and (201) of the hexagonal wurtzite structure ZnO, respectively, which coincides with the given standard JCPDS data card (No. 36–1451). The peaks at 43.93° and 75.30° can be labeled as the (111) and (220) crystalline phases of diamond, respectively, given by the corresponding standard JCPDS data card (No. 06–0675). No other obvious impurity peaks exist. Compared with the n-ZnO/BDD heterojunction, XRD analysis results show that the crystal structure of n-Ce:ZnO NLs has not changed, and keeps a good hexagonal wurtzite structure.

Energy-dispersive X-ray spectroscopy (EDS) spectra are used to evaluate the elemental composition of the n-Ce:ZnO NL/p-BDD heterojunction, as shown in Figure 1e. The spectra present peaks for the main components of the heterojunction, which are only associated with Zn, O, and C elements and a small amount of Ce (the specific elemental composition content is given in the upper right corner), and no other impurity-related peak positions are detected. The quantitative data presented in Figure 1e show that the weight percentages of Zn, O, C, and Ce of Ce:ZnO NLs are 79.36 wt%, 16.09 wt%, 4.36 wt%, and 0.20 wt%, respectively. Hence, the top surfaces of the synthesized Ce-doped ZnO NLs are usually rich in zinc due to the termination with Zn^2+^ [25].

The existence state of Ce ions in the sample is determined by XPS analysis. Full-scan XPS of the n-Ce:ZnO NL/p-BDD heterojunction is shown in Figure 2a. Elemental zinc, oxygen, silicon (mainly originating from the substrate material), and surface carbon (mainly caused by the existence of C element in the diamond substrate and the pollution caused by CO/CO_2_ adsorption on the sample surface) are detected in the spectrum, with no other significant impurities in the sample.

Subsequently, the important elements of Zn(2p), O(1s), and Ce(3d) in the samples are scanned by XPS, and the corresponding high-resolution spectra are obtained (Figure 2b–d) to analyze the apparent chemistry of the n-Ce:ZnO NL/p-BDD heterojunction. Figure 2b shows the high-resolution XPS scan spectrum of Zn2p. The Zn2p signal consists of two symmetrical peaks corresponding to Zn2p_3/2_ at 1021.6 eV and Zn2p_1/2_ at 1044.7 eV. Comparison of the magnitude of the binding energy between the two peaks indicates that the Zn ion exists mainly in the 2-valent form [26]. The signal of O1s can be decomposed into three symmetric peaks by Gaussian fitting (Figure 2c), which are defined as OI1 (530.8 eV), OI2 (532.1 eV), and OI3 (536.3 eV). Among them, the OI1 located at 530.8 eV is mainly related to the O^2-^ ion in the wurtzite structure, which is usually bound by the O^2-^ ion closest to the Zn ion to form a complementary state [27]. The OI2 located at 532.1 eV is associated with the O^2-^ ion in the oxygen-deficient region of the ZnO matrix, which determines the oxygen vacancies inside ZnO. The intensity of this component can reflect the concentration of oxygen vacancies [28]. The OI3 located at 536.3 eV is mainly associated with adsorbed water and oxygen at the grain boundaries or loosely bound oxygen at the sample interface [29]. Figure 2d shows a high-resolution XPS image of Ce3d decomposed into 10 peaks by using a Gaussian function, labeled in order as W1 (881.4 eV), W2 (883.8 eV), W3 (885.9 eV), W4 (888.1 eV), W5 (897.3 eV), W6 (899.9 eV), W7 (901.9 eV), W8 (903.8 eV), W9 (907.3 eV), and W10 (915.6 eV). It is mainly associated with the two sets of spin orbitals of Ce3d_5/2_ and Ce3d_3/2_. The peaks at 888.1, 897.3, 899.9, 901.9, and 915.6 eV can be attributed to the Ce3d_3/2_ and Ce3d_5/2_ photoelectric emissions of CeO_2_ (888.1, 897.3, and 899.9 eV correspond to Ce3d_5/2_ of Ce^4+^; 901.9 and 915.6 eV correspond to Ce3d_3/2_ of Ce^4+^) [30]. The peak position of 888.1 eV suggests that a mixture of Ce (3d^9^4f^2^)O(2p^4^) and Ce(3d^9^4f^0^)O(2p^5^) appears [31]. The peaks at 881.4, 885.9, 903.8, and 907.3 eV are mainly associated with the Ce3d_3/2_ and Ce3d_5/2_ photoelectric emission of Ce^3+^. The peak of 881.4 eV corresponds to Ce3d_5/2_ of Ce^3+^, and the peaks of 885.9, 903.8, and 907.3 eV correspond to Ce3d_3/2_ of Ce^3+^. In addition, the peak of 883.8 eV can be attributed to the metallic phase of Ce [32]. From the above results, the coexistence of Ce^3+^ and Ce^4+^ implies the appearance of oxidized and metallic phases of Ce doped in the ZnO NLs. Ce ions are confirmed to enter the ZnO lattice.

The PL properties of the undoped n-ZnO/p-BDD and n-Ce:ZnO NL/p-BDD heterojunctions excited by 325 nm at RT are shown in Figure 3. Both n-ZnO/p-BDD and n-Ce:ZnO NL/p-BDD heterojunctions exhibit intense UV emission (at 385 and 365 nm, respectively) and blue emission (at 464 and 468 nm, respectively). The n-Ce:ZnO NL/p-BDD heterojunction also exhibits a broader green emission centered at 528 nm. The appearance of the UV emission peak is related to the near band edge emission of ZnO, which is mainly compounded by mutual collisions between free excitons [33]. The blue-green emission of ZnO is mainly associated with defects, such as oxygen vacancies. It should be noted that, here, oxygen vacancies include singly ionized oxygen vacancies (Vo+), doubly ionized oxygen vacancies (Vo++), and neutral oxygen vacancies (Vo), oxygen gaps, zinc vacancies, and zinc gaps (Zni). Among them, the cause of the green emission of ZnO has been controversial, and the most mainstream view is that the combination of photogenerated holes and singly ionized oxygen vacancies (Vo++) in ZnO causes green emission [34]. For the n-ZnO/p-BDD heterojunction, the blue emission peaks at 464 and 469 nm are mainly due to the transition of the zinc gap. Meanwhile, for the n-Ce:ZnO NL/p-BDD heterojunction, the jump between Ce^3+^ and Ce^4+^ and the jump of Ce^3+^ ions from the Zni state to the valence band are involved in the appearance of the blue emission peaks [35]. The emission peak at 483 nm is attributed to the transition between interstitial oxygen and oxygen vacancies [36]. The UV emission peak position of the n-Ce:ZnO NL/p-BDD heterojunction shifts from 385 nm to 365 nm, and the change in UV emission peak position is thought to be caused by the doping of Ce ions in ZnO nanoparticles. The increase in the overall luminescence intensity of the n-Ce:ZnO NL/p-BDD heterojunction indicates that the Ce-doped ZnO has a high electron–hole complexation rate. Figure 3b plots the CIE color coordinates of the n-ZnO/p-BDD and n-Ce:ZnO NL/p-BDD heterojunctions. The chromaticity coordinates of the n-ZnO/p-BDD heterojunction change significantly in the visible region after doping with Ce, shifting from (0.1418, 0.2385) to (0.1413, 0.1853). The emission intensity is also enhanced, reaching from the light blue region to the dark blue region. These results indicate that the n-Ce:ZnO/p-BDD heterojunction has good prospects for application in the dark blue region of light-emitting devices.

Figure 4 (top inset) displays the device structure view of the n-Ce:ZnO NL/p-BDD heterojunction. The positive and negative terminals of the device consist of wires leading from the Ag contacts on the BDD and ITO sides, respectively. The linear *I–V* characteristics between the ITO/Ag and BDD/Ag contacts show a linear relationship of an ohmic contact (Figure 4 bottom inset). The ITO-ZnO also has ohmic contact characteristics, and the work function is nearly the same [37]. The physical parameters of the p-type BDD are measured in accordance with the Hall effect. The carrier mobility, resistivity, and carrier concentration are 38.9 cm^2^ V^−1^ s^−1^, 1.09 × 10^−1^ Ω cm, and 1.46 × 10^18^ cm^−3^, respectively.

Figure 4 shows the current–voltage (*I–V*) characteristics of the n-Ce:ZnO NL/p-BDD heterojunction at various temperatures of 25 °C, 100 °C, and 175 °C. The heterojunction exhibits typical pn diode behavior in terms of *I–V* characteristics, and all curves exhibit typical rectification characteristics with rectification ratios of 1.38, 1.29, and 29.37 at ±8 V at 25 °C, 100 °C, and 175 °C, respectively. The change in rectification ratio is usually directly related to the change in leakage current [38]. When the bias voltage is higher than 4.6 V, the forward currents of the heterojunctions at 100 °C and 175 °C tend to decrease. The turn-on voltage of the heterojunction is 0.6 V at 25 °C, while it decreases to 0.5 V (100 °C) and 0.4 V (175 °C) at higher temperatures. The *I–V* characteristics of the n-Ce:ZnO NL/p-BDD heterojunction demonstrate favorable electrical performance with standard rectification characteristics and low turn-on voltage at the high temperature of 175 °C. The reason is probably that the regular normal diffusion current and excess current are injected for the heterojunction at higher temperatures. These turn-on voltages are much lower than our previous work reported for the undoped n-ZnO/p-BDD heterojunction [14,15], which responds to the fact that the doping of Ce improves the electrical performance of the heterojunction from the n-side. The results show that the n-Ce:ZnO NL/p-BDD heterojunction exhibits more excellent rectification characteristics and electrical performance at the high temperature of 175 °C, which means that the n-Ce:ZnO NL/p-BDD heterojunction is more suitable for high-temperature operation.

To explore the mechanism of the variation trend of *I–V* characteristics, the energy band diagrams of the heterojunction at various temperatures are established, as shown in Figure 5. Owing to the Ce element doping and small thickness (7.9 nm) of the ZnO NLs, the morphology presents a large surface area-to-volume ratio and a great quantity of free surface and oxygen vacancy [39], leading the Ce-doped ZnO NLs to the heavy doping of degenerated semiconductors. For the degenerated n-ZnO, its Fermi level enters the conduction band, and the shallow oxygen vacancy energy level expands to the defect band close to the conduction band (the defect band contains a large number of oxygen vacancies). There exists a common band of energies, in which there are filled states on the n-ZnO. Given that the BDD is nondegenerated, the Fermi level exists in the bandgap. When n-degenerated ZnO is bonded to p-nondegenerated diamond, the injected current is mainly controlled by the holes at the valence band on the diamond side because the barrier height at the conduction band (∆E_c_) is much larger than that at the valence band (∆E_v_). At RT, when a bias voltage is applied, the holes will diffuse to the interface of the heterojunction and can tunnel from the valence band of p-BDD to the common band of the n-Ce-doped ZnO semiconductor. The large band-to-band tunneling current dominates the heterojunction, which results in large forward and reverse currents for the heterojunctions at RT. At high temperature, the Fermi energy level on the p-type diamond side moves and gradually approaches the centerline of the forbidden band, and the bands are uncrossed. Neither n-Ce-doped ZnO nor p-BDD has available empty energy states, resulting in a gradual decrease in tunneling current. With the further increase in voltage, the natural diffusion current and excess current state dominate. In addition, the doping of Ce increases the concentration of oxygen vacancies on the ZnO side, which captures the holes in the excited state and hinders the complexation of electrons and holes, thereby indirectly affecting the conversion between the filled state and the empty energy state on the side of ZnO or diamond and leading to a further decrease in the tunneling current to zero. Therefore, although more thermal emission carriers are generated and injected at high temperatures, as the tunneling current and the carrier complexation decrease, the total current tends to the valley current. Therefore, the *I–V* characteristic curve at a high temperature of 175 °C shows a gradually decreasing trend and exhibits more excellent rectification characteristics.

The ln*I–V* characteristic curves at various temperatures are plotted in Figure 6. The ideal factor (*n*) of the heterojunction is fitted using the following standard diode equation:(1)$$I=I_{s}\left[\exp\left(\frac{qV}{nkT}\right)-1\right],$$ where *I_s_* is the reverse saturation current, *V* is the applied voltage, *q* is the charge, *k* is the Boltzmann constant, and *T* is the absolute temperature. Given that the turn-on voltage of this heterojunction is mainly in the range of 0.4–0.6 V, the fitted voltage interval is limited to 0–0.6 V in accordance with the linear part of the curve, and the ideal factor at various temperatures is calculated. The ideal factors (*n*) calculated at 25 °C, 100 °C, and 175 °C are 6.72, 5.43, and 4.61, respectively. In general, the ideal value of an ideal diode is about 2 [40]. At lower temperatures (25 °C), the n value much larger than 2 is due to structural defects near the interface in the heterojunction, lots of surface states, parasitic rectification junctions, or tunneling effects [41]. Although the value of *n* at RT is not particularly ideal, it is better than the reported heterojunction related to the reported undoped ZnO and Al-doped ZnO heterojunctions [15,42]. When the temperature rises to a higher level, the thermally activated carriers fill most of the defect traps, and the enhanced production–recombination process in the depletion region directly or indirectly leads to the decrease in *n* value [43,44].

The log*I*–log*V* characteristics of the heterojunction at various temperatures are shown in Figure 7. In accordance with the applied bias voltage, the three temperature plots are classified into three different regions. In the low-voltage region (region I), the fitted *I–V* characteristics at 25 °C, 100 °C, and 175 °C follow the power exponential law with *I–V*^1.17^, *I–V*^1.22^, and *I–V*^1.11^, respectively. The exponents are all close to 1, indicating the ohmic behavior of the region. At moderate forward voltage (region II), because of the compound tunneling mechanism of wide-bandgap semiconductor heterojunction devices, the forward current follows the *I*–exp(*α*V) relationship. The fitted injection efficiencies α are obtained in the order of 0.50 (25 °C), 0.52 (100 °C), and 0.57 (175 °C), and these values are located in the semiconductor junction (0.45–1.5 V^−1^) range [45]. The slight increase in α values with increasing temperature indicates improved carrier injection and the appearance of more thermally excited carriers. At higher forward voltages (region III), the fitted *I–V* characteristics at 25 °C, 100 °C, and 175 °C continue to follow the power exponential law, which are *I–V*^2.2^, *I–V*^1.2^, and *I–V*^1.4^, respectively. At low temperatures (25 °C), the curve exhibits a power exponential relationship of *I*∝*V*^2^, which is associated with the space charge-limited current (SCLC) mechanism. The trap-limited SCLC conduction model can explain exponents greater than 2 [46]. Typically, the high density trap states from oxygen surface defects in the n-Ce:ZnO NL/p-diamond heterojunction could capture carriers at low temperatures. At high temperatures (100 °C and 175 °C), the curve exhibits a linear relationship with ohmic behavior because the absence of available states relative to the filled state, with an exponent value of about 1. At this point, the thermal emission carriers no longer flow and tunnel from p-BDD valence band side to the oxygen defect level common band of n-Ce:ZnO NLs.

A plot of ln(*I/V^2^*) versus *1/V* is given in Figure 8 to investigate the interfacial carrier migration mechanism in heterojunctions. In general, the carrier migration mechanisms in heterojunction interfaces can be classified into three categories [47]. (I) When the external temperature is high, lots of carriers cross the interface through the energy generated by thermal excitation to overcome the interfacial potential barrier, and the carrier transport mechanism is mainly thermionic emission, which could be expressed as
(2)$$I=AA^{*}\exp\left[\frac{-\left(\phi_{b}-\sqrt{q^{3}V/4\pi \varepsilon_{0}\varepsilon_{r}d}\right)}{kT}\right],$$
where *A* is the area of the heterojunction, *A** is the Richardson constant, *ε_0_* is the dielectric constant of the vacuum, *ε_r_* is the dielectric constant of the semiconductor, *ϕ_b_* is the height of the potential barrier at *T* = 0 K, and *d* is the height of the potential barrier at the interface. Therefore, the hot electron emission is mainly related to the height of the potential barrier at the interface. The higher the potential barrier at the interface is, the higher the required excitation temperature will be. When the temperature is low, the carrier cannot overcome the barrier caused by thermal energy, so the carrier migration at this time mainly depends on the tunneling mechanism at the interface barrier, which is consistent with the energy band diagram analysis of the heterojunction. (II) When a lower voltage is given, the interfacial carrier transport mechanism at this time is mainly direct tunneling, which can be expressed as
(3)$$\ln\left(\frac{I}{V^{2}}\right)\propto \ln\left(\frac{1}{V}\right)-\frac{4\pi d\sqrt{2m\phi_{b}}}{h},$$
where *h* is Planck’s constant, and *m* is the effective mass of the charge carrier. (III) When higher voltage is given, the interfacial carrier transport mechanism at this time is mainly Fowler–Nordheim (F–N) tunneling, which could be expressed as [48]
(4)$$\ln\left(\frac{I}{V^{2}}\right)\propto -\frac{1}{V}\left(\frac{8\pi d\sqrt{2m\phi_{b}^{3}}}{h}\right).$$

Thus, the conduction of thermal ion emission is mainly temperature dependent, while the direct and F–N tunneling are mainly voltage dependent. At a lower temperature (25 °C), the curve shows a more distinctive feature, namely, the appearance of a voltage inflection point (*V_t_*). The appearance of this inflection point indirectly indicates the presence of direct and F–N tunneling. When *V_t_* > 4 V, *1/V* shows a negative slope trend, signifying the appearance of the F–N tunneling effect. When *V_t_* < 4 V, the electrical transport properties change, and *1/V* takes a logarithmic form, indicating the presence of the direct tunneling effect. At higher temperatures (100 °C and 175 °C), the curve changes significantly, and the inflection point disappears. Therefore, the carriers can generate enough energy to pass through the potential barrier region and be injected by the common nondegenerated pn heterojunction without tunneling current at high temperature.

## 3. Materials and Methods

BDD films were prepared by hot filament chemical vapor deposition (HFVVD) with a thickness of about 4 μm [49]. Ce:ZnO NLs were synthesized using a hydrothermal method. A ZnO seed crystal layer of approximately 20 nm was prepared on the BDD films through magnetron sputtering. The precursor solution for growing Ce-doped ZnO nanomaterials was prepared with 0.2 M zinc acetate dihydrate (Zn(CH_3_COO)_2_ · 2H_2_O), 11 mM cerium nitrate hexahydrate (Ce(NO_3_)_2_ · 6H_2_O), 3 mM hexamethylenetetramine (CH_2_)_6_N_4_), and a certain amount of anhydrous ethanol. Then, the drug was stirred thoroughly with a magnetic stirrer to dissolve rapidly. During the stirring process, a proper amount of NaOH was added to make the pH of the precursor solution 10. After the drug was fully dissolved, the prepared precursor solution was transferred to an autoclave and treated at 150 °C for 24 h. After the reaction, it was rinsed repeatedly with an absolute ethanol solution for 5 min and dried at RT.

The morphology of the samples was examined using scanning electron microscopy (SEM, Carl Zeiss, Oberkochen, Germany). The elemental composition of the samples was analyzed by energy-dispersive X-ray spectroscopy (EDS, Carl Zeiss, Oberkochen, Germany). The phase structure and phase purity of the samples were examined using X-ray diffractometry with Cu Kα radiation (XRD, Rigaku SmartLab, Tokyo, Japan). PL properties were characterized using an FLS920 spectro-fluorophotometer (Edinburgh Instruments, Edinburgh, UK). The *I-V* performance of the heterojunction was measured using a Keithley 2400 source (Keithley Instrument, Cleveland, OH, USA).

## 4. Conclusions

We have successfully grown the Ce-doped ZnO NLs on p-type BDD films using a hydrothermal method to fabricate the n-Ce:ZnO NL/p-BDD heterojunctions. The PL properties and electrical properties and carrier transport behavior at high temperatures of the heterojunctions were investigated. The PL results show that the doping of Ce enhances the defect concentration inside ZnO, which improves the luminescence intensity and causes a significant blue shift of the UV emission peak. The prepared devices exhibit thermal stability and more excellent rectification characteristics and electrical performance at high temperatures. The ideal factor and turn-on voltage decrease with the increase in temperature. The temperature-dependent carrier transport behavior of the n-Ce:ZnO NL/p-BDD heterojunction devices was investigated in depth via energy band diagrams and semiconductor properties. This study extends the design and application of BDD-related heterojunction systems and provides essential insight into the relevant interfacial carrier injection mechanism of the doped metal oxide/diamond heterojunction at high temperatures and in other harsh environments.

## Figures and Tables

**Figure 1 nanomaterials-12-03773-f001:**
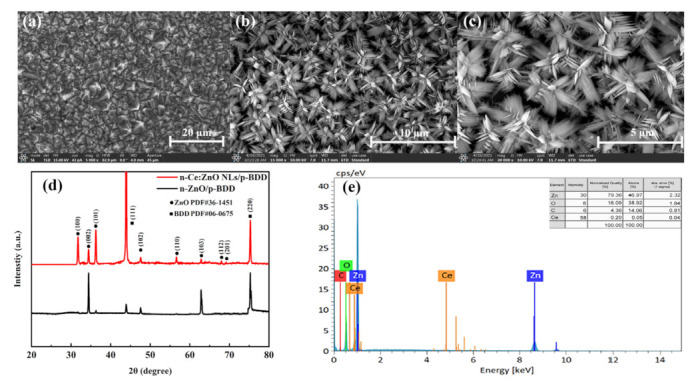
(**a**) SEM of the BDD substrate. (**b**,**c**) SEM of Ce:ZnO NLs deposited on the BDD substrate, respectively. (**d**) XRD spectrum of the n-Ce:ZnO NL/p-BDD heterojunction and n-ZnO/p-BDD heterojunction. (**e**) EDS spectra of the n-Ce:ZnO NL/p-BDD heterojunction.

**Figure 2 nanomaterials-12-03773-f002:**
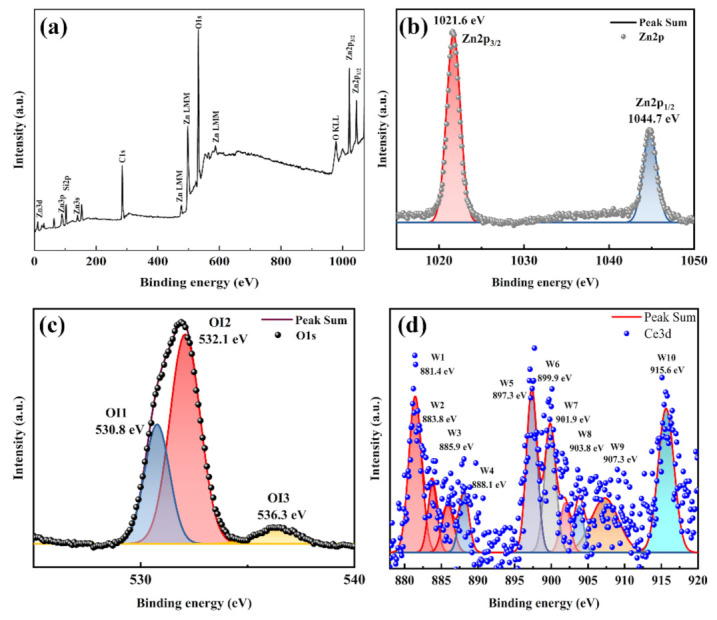
XPS analysis of the n-Ce:ZnO NL/p-BDD heterojunction. (**a**) Full-scan XPS of the n-Ce:ZnO NL/p-BDD heterojunction. High-resolution XPS spectra of (**b**) Zn2p, (**c**) O1s, and (**d**) Ce3d.

**Figure 3 nanomaterials-12-03773-f003:**
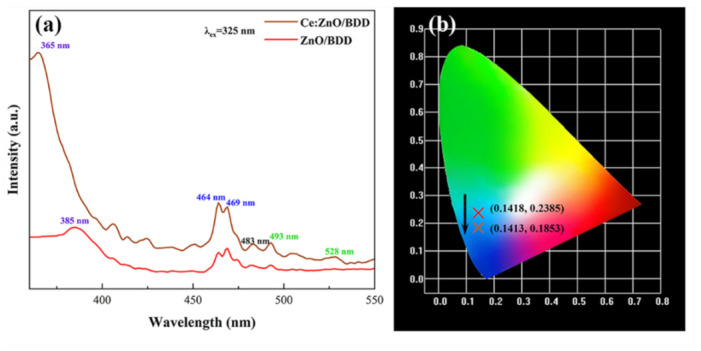
(**a**) PL spectra and (**b**) standard CIE color graphs of the n-ZnO/p-BDD and n-Ce:ZnO NL/p-BDD heterojunctions.

**Figure 4 nanomaterials-12-03773-f004:**
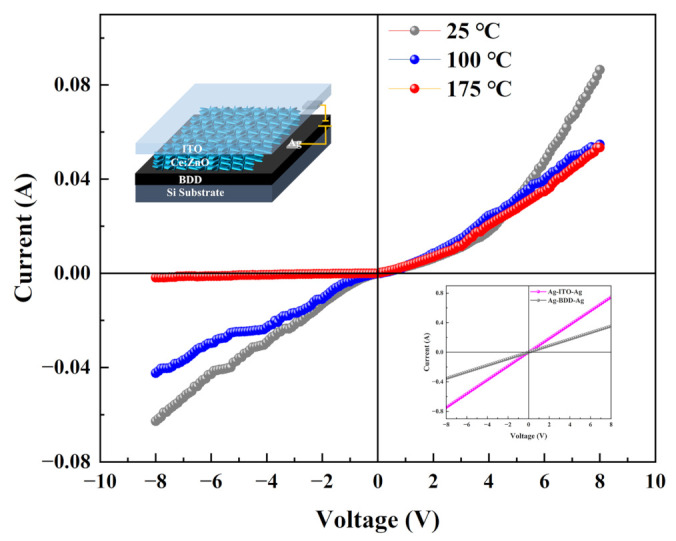
Current–voltage (*I–V*) characteristics of the n-Ce:ZnO NL/p-BDD heterojunction at various temperatures. The top left inset shows the device structure schematic. The lower right inset shows the ohmic contact test of Ag/ITO/Ag and Ag/BDD/Ag.

**Figure 5 nanomaterials-12-03773-f005:**
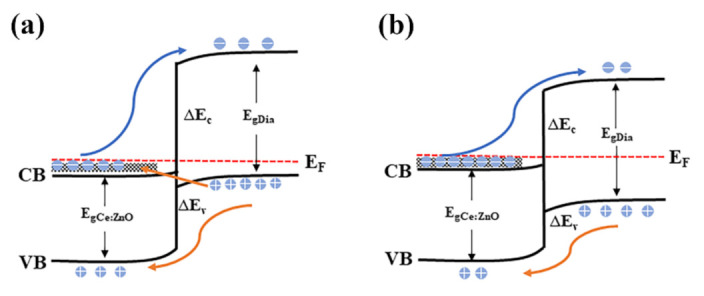
Schematic of the energy band structure of the n-Ce:ZnO/p-BDD heterojunction at various temperatures. (**a**) Room temperature, (**b**) high temperature.

**Figure 6 nanomaterials-12-03773-f006:**
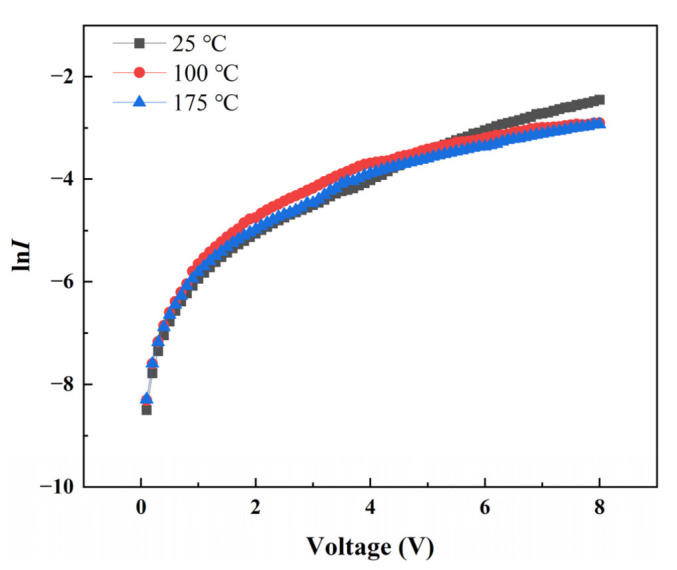
Semilogarithmic current–voltage characteristics of the heterojunction at various temperatures.

**Figure 7 nanomaterials-12-03773-f007:**
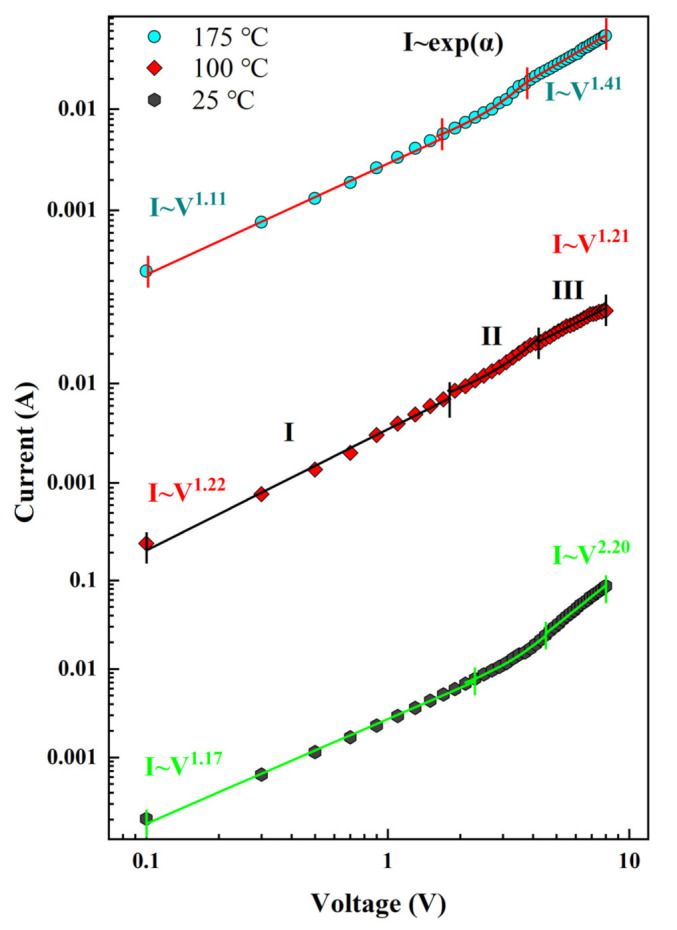
Log*I*–log*V* plots of the n-Ce:ZnO NL/p-BDD heterojunction at various temperatures.

**Figure 8 nanomaterials-12-03773-f008:**
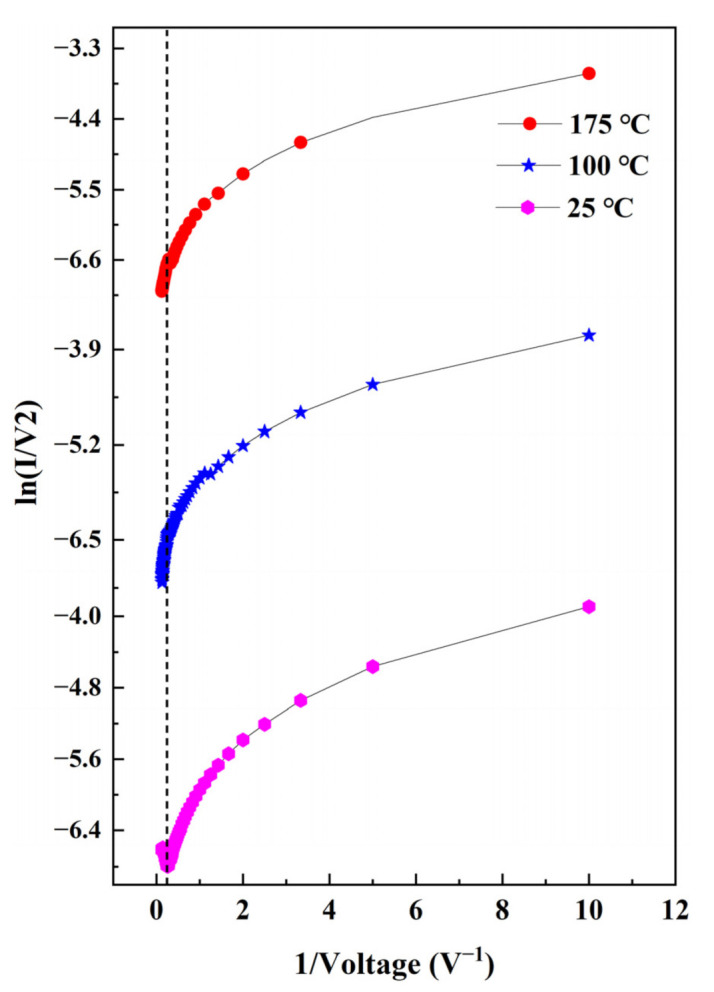
ln(*I/V^2^*) vs. *1/V* plot of the n-Ce:ZnO NL/p-BDD heterojunction at various temperatures.

## Data Availability

The data presented in this study are contained within the article.

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
