# Peer review of "Photoluminescence and Electrical Properties of n-Ce-Doped ZnO Nanoleaf/p-Diamond Heterojunction"

_nanomaterials, 2022, doi:10.3390/nano12213773_

Round 1
Reviewer 1 Report
This paper demonstrates the effect of degenerated doping of ZnO with Ce to obtain performance enhancement in n-Ce-ZnO and p-diamond heterostructure. Detailed analysis of the chemical composition and the binding energy of n-ZnO, n-Ce-ZnO, and p-BDD have been presented using SEM, XRD, EDX, and XPS. Further, the influence of the temperature on the rectification behavior of heterojunction has been delivered with detailed scientific analysis. Overall, the paper in its current form is looking good for publication.
Author Response
We are grateful for the reviewer's recognition.
Reviewer 2 Report
In this manuscript, the authors synthesized the Ce-dopped ZnO and studied electrical properties of Ce-ZnO/BDD films. The IV characteristic depending on temperature of the Ce-ZnO/BDD heterojunction device was systematically studied. I can suggest its publication in the journal after considering following comments.
1. The authors claimed that the PL intensity of Ce-ZnO is higher that that of ZnO. I am wondering whether PL characterization was measured at the same condition such as film thickness. Please give QY of Ce-ZnO and ZnO in solution state.
2. Figure caption of Fig1 bc need to be clear.
3. In figure 1d, Please provide ZnO XRD spectra of before and after Ce doping in figure 1d.
4. The XPS spectra in Fig 2d is not clear and looks like as a noise.
Reviewer 3 Report
In this work, the authors have successfully grown the Ce-doped ZnO NLs on p-type BDD films using a hydrothermal method to fabricate the n-Ce:ZnO NLs/p-BDD heterojunctions. The PL properties, and electrical properties and carrier transport behavior at high temperatures of the heterojunctions were investigated. The prepared devices exhibit thermal stability and more excellent rectification characteristics, electrical performance at high temperatures. Contents are interesting, and I can recommend a publication after taking into account below comments.
1. Figure 1 (d) and (e): Improve the quality of drawings. The captions on the drawings are not readable.
2. Figure 2 (d): Explain why you describe the data in this way. You can find many more bands in this spectrum. Why limited to just those? And was it even possible to decompose such a spectrum into Gaussians?
Round 2
Reviewer 3 Report
Thanks for answers.